# Neural Operator Learning for Ultrasound Tomography Inversion

**Haocheng Dai**[*1]                                                    HDAI@SCI.UTAH.EDU
**Michael Penwarden**[*1]                                MPENWARDEN@SCI.UTAH.EDU
**Robert M. Kirby**[1]                                              KIRBY@SCI.UTAH.EDU
**Sarang Joshi**[1]                                                  SJOSHI@SCI.UTAH.EDU

[1] *Scientific Computing and Imaging Institute, University of Utah, Salt Lake City, UT*

## Abstract

Neural operator learning as a means of mapping between complex function spaces has garnered significant attention in the field of computational science and engineering (CS&E). In this paper, we apply neural operator learning to the time-of-flight ultrasound computed tomography (USCT) problem. We learn the mapping between time-of-flight (TOF) data and the heterogeneous sound speed field using a full-wave solver to generate the training data. This novel application of operator learning circumnavigates the need to solve the computationally intensive iterative inverse problem. The operator learns the non-linear mapping offline and predicts the heterogeneous sound field with a single forward pass through the model. This is the first time operator learning has been used for ultrasound tomography and is the first step in potential real-time predictions of soft tissue distribution for tumor identification in beast imaging. The project is available at https://github.com/aarentai/Ultrasound-Tfno-MIDL.

**Keywords:** Neural Operator, Ultrasound Tomography, Inversion, Breast Imaging

## 1. Introduction

The goal of ultrasound tomography is to estimate the acoustics properties of an object using the transmission of sound waves (Javaherian and Cox, 2021). In this paper, we focus on the breast imaging problem to serve as a test case for applying neural operator learning to ultrasound tomography. Ultrasound tomography to obtain cross-sectional images of breast structures is a promising non-ionizing imaging modality for the diagnosis and screening of breast cancer (Guo et al., 2018). Numerous detector geometries have been proposed for speed-of-sound ultrasound computed tomography (Malik et al., 2018). Although the methodology proposed is applicable to all the geometries, we focus on the ring transducer geometry (Sandhu et al., 2015).

Neural operators learn mappings between infinite-dimensional function spaces that are discretization-invariant and have universal approximation properties (Kovachki et al., 2021). In particular, Fourier neural operators (FNOs) have been shown to have high accuracy, low inference time, and robustness to noise and generalization error when learning the solution operators of partial differential equations (PDEs) (Li et al., 2020). In this paper, we exploit the fact that the method learns general function-to-function maps, not limited to PDE solutions, and apply it to ultrasound tomography inversion.

[*] Contributed equally. H. Dai and S. Joshi were supported by NSF grant DMS-1912030 and NIH grant 1R01CA259686. M. Penwarden and R. M. Kirby were supported by AFOSR MURI grant FA9550-20-1-0358.

## 2. Methodology

We utilize a tensorized Fourier neural operator (T-FNO) to learn the mapping between the 2D emitter by receiver time-of-flight (TOF) field and the spatial 2D sound speed (SS) field. The T-FNO model features 7.3 million learnable parameters, with 64 modes, 32 hidden channels, and 32 projection channels. We provide comparisons with a standard U-Net (Ronneberger et al., 2015), highlighting improved generalization predictive capabilities in the output space with the T-FNO. The U-Net lifts the input field to 32 hidden channels and has 7.7 million parameters – 0.4 million more than the T-FNO. Hyperparameter tuning was performed to determine model sizes with sufficient expressivity without being over-parameterized. Adam optimization was performed for 10,000 epochs, which took 1 to 2 hours on a single Nvidia Titan RTX, with an initial learning rate of $1 \times 10^{-3}$. Additionally, a `ReduceLROnPlateau` scheduler was adopted with a 0.5 learning rate reduction factor.

To generate synthetic training and testing samples, Gaussian random fields (GRFs) were used to emulate the variations in soft tissue and binned into binary values (1450, 1550 m/s). The outside of the desired synthetic breast tissue region is replaced with water-like SS values (1500 m/s) and "skin" (1580 m/s) is added around the breast tissue. This results in randomly generated fields, as shown in the second column of Figure 1, which is ideal for synthetically creating a large dataset. The fidelity of both input and output fields is $128 \times 128$.

A *k-space pseudo-spectral* method (k-Wave) MATLAB package (Treeby and Cox, 2010) was used to run a full-wave numerical forward simulation over the randomly generated heterogeneous SS fields and a homogeneous density field, given an equally distributed set of 128 emitter locations and 128 receivers. We used a Daubechies 8 wavelet as the time-varying excitation pulse for all emitters, emulating a physical transducer. The TOF was determined using cross-correlation between the emitter and receiver signals. The discrepancy is taken between the TOF of the breast inside a water bath and only water, resulting in fields that highly correlate to the variation in SS fields. The TOF discrepancy and SS fields were then min-max normalized before training, and an 80:20 train-test split was used.

## 3. Results

Table 1: Comparison between T-FNO and U-Net on the full dataset inversion problem

| Model | GRF Correlation | Noise | Testing MSE | Training MSE |
|---|---|---|---|---|
| T-FNO | High | Clean | $\mathbf{2.01 \pm 0.33 \times 10^{-2}}$ | $0.69 \pm 0.11 \times 10^{-2}$ |
| | | 10% | $\mathbf{2.06 \pm 0.34 \times 10^{-2}}$ | $0.68 \pm 0.11 \times 10^{-2}$ |
| | Low | Clean | $\mathbf{3.27 \pm 0.16 \times 10^{-2}}$ | $1.00 \pm 0.06 \times 10^{-2}$ |
| | | 10% | $\mathbf{2.67 \pm 0.17 \times 10^{-2}}$ | $1.53 \pm 0.08 \times 10^{-2}$ |
| U-Net | High | Clean | $2.52 \pm 0.44 \times 10^{-2}$ | $\mathbf{0.12 \pm 0.03 \times 10^{-2}}$ |
| | | 10% | $2.79 \pm 0.42 \times 10^{-2}$ | $\mathbf{0.01 \pm 0.01 \times 10^{-2}}$ |
| | Low | Clean | $3.81 \pm 0.17 \times 10^{-2}$ | $\mathbf{0.42 \pm 0.05 \times 10^{-2}}$ |
| | | 10% | $4.02 \pm 0.21 \times 10^{-2}$ | $\mathbf{0.02 \pm 0.01 \times 10^{-3}}$ |

In Table 1, the mean squared error (MSE) over the full dataset is reported for a variety of setups. White noise was added to the receiver time series prior to cross-correlation to assess robustness. We observe that the T-FNO outperforms the U-Net at test time under

all conditions, whereas the U-Net better fits the training set but does not generalize well. A single inference with the T-FNO takes 1.44 seconds compared to 1.57 with the U-Net. Higher correlation fields prove easier to infer for both models, although they incur greater variance in the errors.

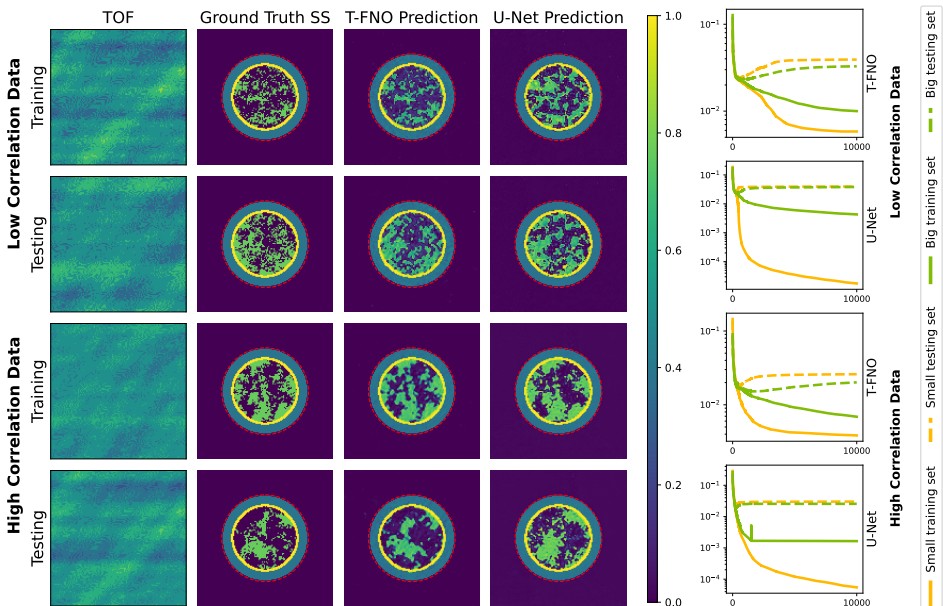

Figure 1: Single realization of train/test set examples. The dashed red line indicates the location of the evenly distributed emitters and receivers. The region outside the transducer ring is masked out for improved training and quantity of interest comparison but is present in the full-wave simulation to mitigate reflection.

In Figure 1, the respective models were trained to learn the mapping between the TOF discrepancy (first column) and the ground truth SS (second column) with resulting SS predictions (third and fourth column) for one realization in the dataset. The loss convergence plots are shown for differing dataset sizes of 100 (small dataset) and 200 (big dataset), respectively (fifth column). Empirically, we observe that the T-FNO better captures the overall trends in the data, while the U-Net is prone to overfit the training data. This is also shown in the loss convergence plots, in which the U-Net suffers from considerable generalization error, even when additional examples were provided.

## 4. Conclusion

We have proposed using neural operators to accurately and efficiently solve the full-wave inverse problem on synthetic ultrasound tomography. Our novel application of the T-FNO improves over the baseline U-Net, laying the foundation for real-time accurate predictions of soft tissue distribution for tumor identification on breast imaging. Additionally, the application of both U-Net and T-FNO to this problem formulation is itself novel since both are real-time predictors and do not require computationally expensive ray-based inversion once trained. Future work will explore solving the inverse Helmholtz problem instead of the full-wave solution for sound speed reconstruction as well as training and testing on non-synthetic real breast image data.

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
