# OpenReview forum: "Neural Operator Learning for Ultrasound Tomography Inversion"
_MIDL.io/2023/Short_Paper_Track — MIDL 2023 Short paper track Poster_

### Official Review · Reviewer_5ozm · 2023-04-24
**This paper proposes to utilize the developed Fourier neural operators (FNO) to learn the mapping function between the 2D emitter by receiver time-of-flight (TOF) field and the spatial 2D sound speed (SS) field. The authors compared the proposed method with one of the widely used neural networks - Unet.**

**Rating:** 4
**Confidence:** 5

**Review:**

Strength:

-The proposed idea of learning the mapping function, rather than treating the problem as a regression task in the network is interesting.

-The paper is well organized and easy for readers to follow.

Weaknesses:

-While the proposed idea is interesting, there are many critical details/questions missing in the current manuscript. First of all, the original FNO is learned as a surrogate to solve PDEs/ODEs. The FNO network requires the use of the numeral solution to the PDEs/ODEs  as “ground truth” in the training process. This is not clear in the proposed methodology. More details are needed for the readers to fully understand the proposed work. Second, the original FNO has a recurrent neural network after being lifted, which makes the computational cost (including time and memory) of training FNO high. While the authors mentioned that the proposed T-FNO is 0.4 millions less, I wonder whether the memory consumption is also lower compared to Unet (that down-samples data at each layer of the network).

-The experimental validation is not sufficiently convincing. First of all,  there are many other advanced networks that may have better performance than Unet, for example, transformer or diffusion models. Missing comparisons with these state-of-the-art is less convincing. Second, the experimental results only show 10% noise, which is relatively low. What happens if the noise is higher beyond this number?

-The T-FNO prediction in Fig. 1 is more blurry than Unet with less sharp details. The authors need to provide the insights behind this result. Is it because of the predicted mapping function of T-FNO is overly smooth, or the network is less efficient to capture details with high-resolutions?

---

### Official Review · Reviewer_1xxm · 2023-04-25
**Interesting application of neural operator learning to ultrasound**

**Rating:** 9
**Confidence:** 4

**Review:**

This paper proposes to apply neural operator learning usually used to learn mapping in PDE solvers to the problem of tomographic reconstruction. The authors train the model on simulated data and evaluate its performance, showing improvements over a UNet model with more parameters. This will make a good contribution to the conference; neural operators seem like an interesting direction for image reconstruction broadly, beyond the application considered here.